# Morphological Peculiarities of the Pelvic Autonomic Nervous System and Their Impact on Clinical Interventions in the Lesser Pelvic Region

**DOI:** 10.3390/medicina59010072

**Published:** 2022-12-29

**Authors:** Roman Kuruc, Andrea Szórádová, Jarmila Kristová, Martina Solárová, Jozef Šidlo, Viktor Matejčík

**Affiliations:** 1Institute of Forensic Medicine, Faculty of Medicine, Comenius University, Sasinkova 4, 81108 Bratislava, Slovakia; 2Medico-Legal Department of Health Care Surveillance Authority, Antolská 11, 85107 Bratislava, Slovakia; 3Faculty of Nursing and Professional Health Studies, Slovak Medical University, Limbová 12, 83303 Bratislava, Slovakia; 4Department of Neurosurgery, Faculty of Medicine, Comenius University, University Hospital, Limbová 5, 83305 Bratislava, Slovakia

**Keywords:** autonomic nervous system (ANS), anatomical variations, pelvic ANS and plexus, health

## Abstract

*Background*: The aim of the work is to define the morphological peculiarities of the pelvic autonomic nervous system (ANS) and their importance in the clinical and surgical interventions in the lesser pelvis. *Material and methods*: Anatomical variations in the formation of the pelvic ANS were observed in 20 cadavers. The study included 17 men (85%), aged 18 to 84, and 3 women, aged 27 to 86. The average age was 53.8 years. The subjects most often died by violent death in car accidents, by asphyxia, or by sudden death. The study was approved by the Ethics Committee of the Health Care Surveillance Authority, Bratislava, Slovakia. We studied cadavers without congenital or detected anomalies, cancer, deformities of the body, or spinal or abdominal surgeries within 24 h of death. We observed a relationship between the dimensions and the number of ganglia, as well as the number and course of nerve branches and anastomoses. In the pelvic area, we observed the hypogastric plexus superior, hypogastric plexus inferior, and the truncus sympathicus. In all cadavers, we clarified the lumbosacral plexuses after evisceration. In the lumbosacral region, the roots were defined based on their participation in the formation of the plexuses. To show the intimate relationship between both systems, we also focused on the details of the structure (rami communicantes) related to the connections of the ANS with the spinal nervous system. *Results*: Anatomical variations in the formation of the pelvic ANS were observed in all cases. We included cases with more than two truncus sympathicus ganglia as the segmental type. The segmental form occurred in 14 (70%) cases, and was concentrated in 6 (30%) cases. Rami communicantes provided anastomoses to the spinal nerves. Small ganglia were observed on the rami communicantes. With the concentrated type, we observed the division of the sympathetic and parasympathetic systems. With the segmental and concentrated forms, symptoms of the “diffuse form” may occur, which we observed in all cases. We observed significant right-left asymmetry and differences in the formation of ganglia and anastomoses. *Conclusions*: This study allowed us to identify and describe the morphological peculiarities of the pelvic ANS and their possible influence on the clinical picture. Asymmetry and dependence of their occurrence on the type of ANS was observed. The variations were frequent. Their preoperative diagnosis is difficult to impossible. The absence or lack of intraoperative vigilance can lead to the damage of pelvic ANS during operations and blockades of the pelvic plexus. The acquired knowledge can be helpful in clarifying clinical signs and symptoms of these conditions.

## 1. Introduction

The human body exhibits a certain degree of differences and variations, especially in terms of its shape and structure. Anatomical variations are very interesting and thought-provoking for anatomists, pathologists, forensic physicians, and clinicians. There are different types of anatomical variants and peculiarities. They can be frequent, infrequent, or rare (sporadic). Frequent variations can even reach up to 100% occurrence. Many anatomical variants do not require clinical attention, but some may present diagnostic problems or cause adverse clinical symptoms [1].

The autonomic nervous system (ANS) is a division of the peripheral nervous system, the system of nerves and ganglia that innervates the blood vessels, heart, smooth muscles, viscera, and glands, controlling their involuntary functions, consisting of both sympathetic and parasympathetic divisions. The innervation of the human pelvis has been the subject of numerous studies and reports by scientists who study cadavers. Despite the fact that these studies have been conducted over a long period of time, the information regarding the structure, course, and range of innervation of the hypogastric plexuses still remains scant. Knowledge of the structure of the pelvic ANS plexus must be the basis for a correct understanding of ANS disorders. These peculiarities can explain the contradiction and diversity of clinical manifestations. We have come across literature and scientific publications devoted to this issue. Only rarely did we come across manuscripts focused on bilateral morphological deviations in the formation of the pelvic plexus. Anatomical, and especially surgical, textbooks usually do not pay enough attention to the anatomical peculiarities of the formation of the pelvic plexus, which can complicate the solution of many pathologies and consequently, medical and nursing care [2,3,4,5,6,7,8,9,10,11,12,13,14,15,16,17].

The integrity of the pelvic autonomic nervous system is essential for proper bowel, bladder, and sexual function. In pelvic surgery, dysfunctions of both the anatomical structures and sexual functions are known to be caused by iatrogenic lesions of the inferior hypogastric plexus (IHP), as the pelvic ANS is difficult to define and dissect. The inferior hypogastric plexus is a paired plexus of autonomic nerves and ganglia that supplies the viscera of pelvic cavity. Limited visibility in the lesser pelvis may complicate the identification and sparing of the autonomic nerves, and therefore requires the meticulous planning and execution of surgery. The inferior hypogastric plexus lends its clinical relevance to various urogenital pain syndromes, including prostatitis, endometriosis, chronic sacral pain, and rectal pain. Therefore, any surgical procedure in this anatomical area must be addressed with due diligence with respect to the pelvic plexus, which may be inadvertently damaged. Its damage may cause a slew of related postoperative symptoms (digestive symptoms, chronic pelvic pain, and sexual dysfunction), as well as compromise the pelvic autonomic innervation. Interest in the precise anatomy of the pelvic plexus and IHP increased with the introduction of the telerobotic method of treatment, where magnification and perfect lighting allow for easier preservation of the neural structures [3,5,11,12,18,19,20,21,22,23,24].

Regarding this, we conducted the study dealing with morphological anatomical variations of the pelvic ANS bilaterally, in the terms of diagnosis and clinical impact.

## 2. Materials and Methods

Detailed dissections were performed on 20 cadavers. The study included 17 men (85%), aged 18 to 84, and 3 women (15%), aged 27 to 86. The average age was 53.8 years. The subjects most often died by violent death in car accidents, by asphyxia, or by sudden death. The study was approved by the Institutional Ethics, Arbitration, and Disciplinary Committee of the Health Care Surveillance Authority (Approval Code 3/2022). We studied cadavers without congenital or detected anomalies, cancer, deformities of the body, or spinal or abdominal surgeries within 24 h of death. In all of subjects, the pelvic plexus was clarified bilaterally. The body was lying down. Access to the neural structures of the abdominal cavity and pelvis was obtained through a longitudinal incision in the midline, from the jugular fossa to the symphysis. In the pelvic area, we observed the hypogastric plexus superior, hypogastric plexus inferior, and truncus sympathicus. 

After retroperitoneal clarification of the pelvic splanchnic nerves entering the pelvic plexus from the anterior, lateral, and laterocaudal approach, we dissected the pelvic plexus and nerve structures innervating the bladder, prostate, seminal vesicles, and rectum. In women, we dissected the uterus and vagina. By pressing on the lower pole of the scrotum, we pushed the testis and epididymis upwards and visualized the plexus spermaticus and plexus deferentialis. The fascial structures covering the sacral nerve roots were excised. The parasympathetic nerve roots were identified using the distance from the anterior nerve roots S2–S4 and dissected along their course forward to their exit at the lateral border of the rectum. In all cadavers, we clarified the lumbosacral plexuses after evisceration. In the lumbosacral region, the roots were defined based on their participation in the formation of the plexuses. To show the intimate relationship of both systems, we also focused on the details of the structure (rami communicantes) related to the connections of the ANS with the spinal nervous system. Histological techniques (hematoxylin and eosin) were used to reliably identify ganglia obtained during dissections to determine whether the identified tissue structures contained neurons.

## 3. Results

In the lesser pelvis, we observed a relationship between the dimensions and the number of ganglia, as well as the number and course of nerve branches and anastomoses. We observed anatomical variations of the pelvic ANS in all 20 cases. In all cases, there was significant lateral asymmetry (Table 1). According to the number of ganglia, we classified the variations as segmental type (3 or more) and concentrated type (1 or 2). The segmental type was found in 14 (70%) cases, and the concentrated type was found in 6 (30%) cases (Table 2). The segmental type was observed in 12 men and 2 women; the concentrated type was found in 5 men and 1 woman. With the segmental and concentrated form of ANS, symptoms of the so-called diffuse form occurred in all cases. With the diffuse type, we observed transitions between both types of ANS formation. In similar cases, it is not possible to determine from which source the nerve branches arise, as they contain fibers of both systems (sympathetic and parasympathetic) in their composition. 

Histologic analysis confirmed neural tissue in all samples of the peripheral ganglia. Sympathetic and parasympathetic neurons were identified based on the qualitative characteristics of cytoplasmic lipofuscin—neurons of the sympathetic system with eccentrically deposited nuclei and significant content of cytoplasmatic lipofuscin surrounded circumferentially by irregularly arranged satellite cells between the elongated nuclei of the Schwann cells. Neurons of the parasympathetic nerve were formed by large, lipofuscin pigment-depleted pseudounipolar cells.

In 14 cases in the segmental arrangement of the truncus sympathicus ganglia, more than two ganglia of rami communicantes provided anastomoses to the spinal nerves and participated in the formation of the pelvic plexus (Figure 1). In the sacral section, rami communicantes and rami interganglionares were observed more often with small ganglia (Figure 2). In 6 cases in the concentrated type, we observed a decrease in the number of ganglia tr. sympathicus (Figure 3), as well as the absence of small ganglia on the connecting branches. With the concentrated type, we observed not only a small number of junctions, but also a division of the sympathetic and parasympathetic systems.

In men with the segmental type of lesser pelvic organ innervation (Figure 1), the plexuses of the rectum and bladder contained several small ganglia, the connections with tr. sympathicus were numerous, the nn. erigentes pelvici formed connections with the plexus sacralis (4), which were numerous, and the plexus hypogastricus inferior (5) consisted of many branches arising from the plexus hypogastricus superior (6).

In the concentrated type of lesser pelvic organ innervation (Figure 3), the plexuses of the rectum (1) and bladder (2) consisted of a small number large nerve ganglia and the junction with the tr. sympathicus (3) was more weakly expressed than in the previous case (Figure 1). Nn. erigentes pelvici were rare. The plexus hypogastricus inferior was formed by a small number of branches, and the connections between them were less numerous. The other images show the uterovaginal plexus segmental type (Figure 4) and the concentrated type (Figure 5); the plexus spermaticus segmental type (Figure 6), and the concentrated type (Figure 7). 

The other images show the significant asymmetry of the pelvic plexus in men (Figure 8) and asymmetric formation of the uterovaginal plexus in women (Figure 9).

In general, the form and alignment of the pelvic autonomic nerves displayed large individual variations, which could have a clinical implication on the postoperative function of the pelvic organs. From the observation of the extreme forms of formation of the pelvic ANS, it follows that in one section of the partition, as well as in others, ganglia can be scattered or concentrated.

## 4. Discussion

This study is the first description of morphological peculiarities with relation to the pelvic ANS in fresh cadavers. In this work, we provide a bilateral systemic view of the sources and distribution of nerve roots, plexuses, and ganglia, depending on their type. It describes in detail the topography of the inferior hypogastric plexus (IHP), which may be useful in improving nerve-sparing surgical approaches during pelvic surgery. The autonomic nerves of the lesser pelvis are particularly prone to iatrogenic lesions due to their exposed position during manifold surgical interventions. Nevertheless, the cause of rectal and urinary incontinence or sexual dysfunctions remains largely understudied [3,5,18,20,24,25].

We demonstrated a more specific distribution in the participation of different ganglia and ventral rami. Sacral sympathetic ganglion S2 and S4 were the primary sources of parasympathetic fibers to the IHP. Variations in results may be due in large measure to the fact that the autonomic nerve fibers are extremely tiny and difficult to dissect. There are similar sources for the pudendal nerve and the parasympathetic fibers forming the pelvic splanchnic nerves, suggesting that the somatic autonomic nervous system may be interrelated. The anatomical variations of root presentation, as well as possible variations, play an important role in the explanation of clinical symptoms. Significant variations in the formation of pelvic splanchnic nerves and the communication between them were observed. This brings us closer to the correct understanding of non-uniformity and idiosyncrasies in the development and course of pathological processes, differences in the clinical picture of the same diseases in different people, and non-uniform success in operations. The preoperative diagnosis of these variations is difficult. Anatomic variability and the inability to visualize the smaller diameter nerve fibers likely underlines the reasons why some postoperative visceral and sexual dysfunction occurs in spite of careful dissection and adequate surgical techniques. These findings highlight the importance of a discussion with patients about the risks that are associated with interrupting autonomic fibers during the surgical procedure [5,9,12,18,19,20,22,23,26,27].

The differences in and distribution of the ganglia are also reflected in the peculiarities of the connections between the parts of the sympathetic and parasympathetic systems. It is difficult to imagine that the differences in the structure of the pelvic plexus would not be reflected in the nature of the clinical problems. Paths of pain radiation and their localization are different, and it is difficult to determine their specificity for a particular organ. This diversity, from the point of view of radiation, can be explained by differences in the structure of the pelvic plexus. Irritation of the pelvic plexus caused by a painful condition of any organ can spread to different areas of the body, which often do not have a segmental connection with the given organ. The instability of the junctions may explain the instability of these symptoms. The instability of pain and the complexity of the body’s overall pain reactions depend on the individual characteristics of the patient, the nature of the reactions of the nervous system, and in each individual case, on the existing peculiarities of the structures of the ANS as a whole. It is clear that anatomical and physiological diversity and variation is a canon of living organisms. Variations in the nervous system are fewer than in any other organ system. This is undoubtedly due to the fact that the nervous system, more or less, controls the function of all the other organs. The variations in our set were frequent; their preoperative diagnosis is difficult to impossible. We analyzed the formation of the pelvic ANS in 20 fresh cadavers. We observed anatomical variations and significant lateral asymmetry of the pelvic ANS in all cases. In the literature, we did not observe a division according to the number of ganglia. Therefore, according to the number of ganglia, we classified the type of formation into segmental and concentrated types. The concentrated type occurred in 6 (30%) cases, and the segmental type of ANS formation occurred in 14 (70%) cases. In the concentrated type, we observed the division of the sympathetic and parasympathetic system. With the segmental and concentrated form, symptoms of the diffuse form may occur. We observed symptoms of the diffuse form in all cases. We also observed significant lateral asymmetry in the formation of the pelvis splanchnic nerves, as well as differences in the formation of the ganglia, and anastomosis. Anatomical variations of ANS have been described in previous studies. There are certain permissible anatomical variations described in the atlases, but the ones we observed here were more prominent and at the same time, asymmetrical. We did not come across works dealing with pelvic ANS variations bilaterally, but rather, we found papers dealing only with the variations of individual ANS nerves [11,13,15,16,23,28,29,30,31,32,33]. 

In our group, we observed these variations in all cases of the lumbosacral region. Knowledge of the possible variations in the formation of the pelvic plexus and the ANS of the lesser pelvis is useful in solving abdominal pathologies. Nn. splanchnici connect the pelvic plexus with the truncus sympathicus in the lumbosacral region, and through it, and rami communicantes are connected with corresponding segments of the spinal cord; differences in the level of formation of these nerves can clarify the peculiarities of pathological manifestations in diseases of the internal pelvic organs.

## 5. Conclusions

The description of pelvic ANS and its variants described in this cadaver study may be useful when carrying out pelvic surgery in this anatomical region. Awareness of the structure of the IHP might significantly improve outcomes in all fields of pelvic surgery. The absence or lack of perioperative vigilance can lead to damage to the pelvic ANS. We believe that the data obtained from anatomical dissections can be helpful for many surgeons, as well as for nursing practice and rehabilitation. It is necessary to realize that insufficient knowledge, as well as a misunderstanding of the basic pathophysiological mechanisms, can lead to erroneous considerations and the implementation of incorrect medical and nursing procedures which can endanger the health and life of the patient. Patient care in surgery is very demanding and requires both the physician and the nurse to possess adequate knowledge, to use critical thinking, and to engage in close cooperation with all members participating in the provision of health care. 

Several factors influenced our study, especially the relatively small number of observations and the significant difference in the number of men and women in the study population. These are the limits affecting our interpretation and the possibility of generalizing our findings.

## Figures and Tables

**Figure 1 medicina-59-00072-f001:**
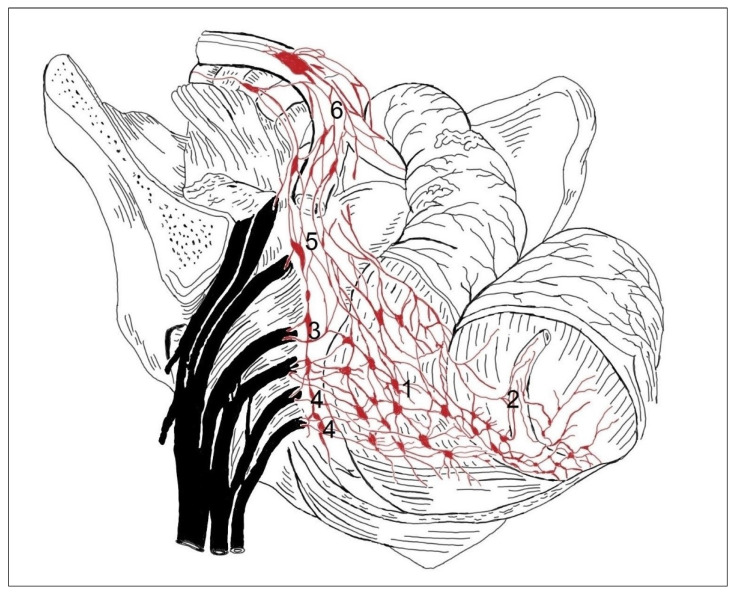
In the segmental type of lesser pelvic organ innervation in men, the rectal (1) and vesical (2) plexuses contain many small ganglia. Multiple connections with the sympathetic trunk are present (3). Nn. erigentes (pelvici) form connections with branches of the sacral plexus (4), which are also numerous. The inferior hypogastric plexus (5) consists of multiple nerve branches arising from the superior hypogastric plexus (6).

**Figure 2 medicina-59-00072-f002:**
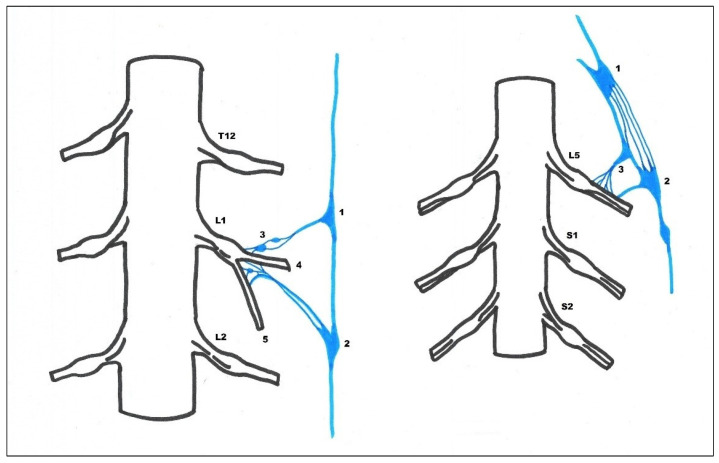
(**Left**): Between the penultimate and last ganglia of the lumbar part of the sympathetic trunk, multiple interganglionic connections are seen. The first (1) and the second (2) sympathetic trunk ganglion is connected by multiple rami communicantes with the L1 nerve, with additional small ganglia (3) near the intervertebral ganglia along the thin communicating branches. In the area of the intervertebral ganglia, the sympathetic branches are associated with the dorsal branch of the spinal nerve (r. dorsalis nervi spinalis) (5), as well as with its ventral branch (r. ventralis) (4). In some cases, an increase in the number of ganglia in the sympathetic trunk is combined with multiple interganglionic branches (rr. interganglionares), along which other small ganglia occur. (**Right**): In the following picture, multiple interganglial connections are visible between the last one (1) and last two (2) ganglia of the lumbar part of the sympathetic trunk. On one fascicle, there is a relatively large ganglion (3), which, like the last lumbar, is connected by rami communicantes with the spinal nerve L5.

**Figure 3 medicina-59-00072-f003:**
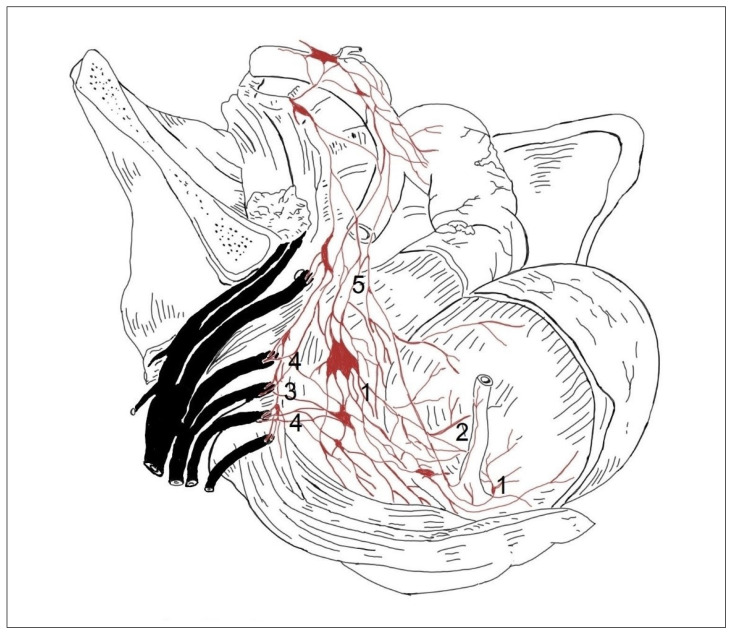
In the concentric type of sympathetic trunk, the plexuses of the rectum (1) and the urinary bladder (2) consist of a small number of large nerve ganglia, and the connections with the sympathetic trunk (3) are more weakly expressed. Nn. erigentes pelvici (4) were rare. The plexus hypogastricus inferior (5) was formed by a small number of branches, and the connections between them were less numerous.

**Figure 4 medicina-59-00072-f004:**
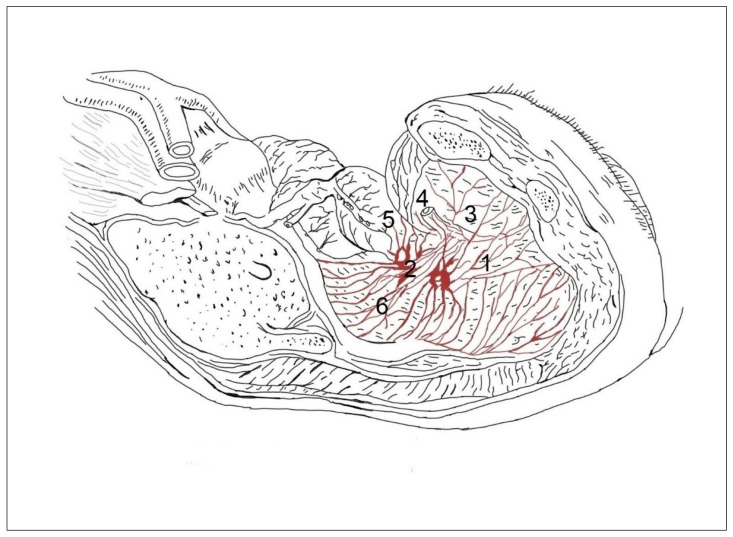
Innervation of lesser pelvis organs in women. In the segmental type, the uterovaginal plexus is formed from the small ganglia (2), from which arise separated branches to the outer surface of the vagina (1), to the urinary bladder (3), ureter (4), vagina (5), uterus (6), and rectum (7).

**Figure 5 medicina-59-00072-f005:**
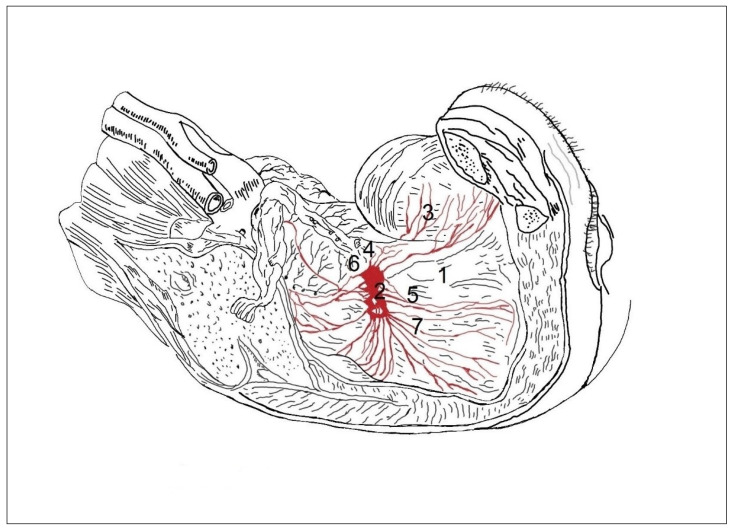
In the concentric type, 1 to 2 large-sized ganglia (2) are observed, from which arise separated branches (almost unconnected nerves on the periphery) to the outer surface of the vagina (1), to the urinary bladder (3), ureter (4), vagina (5), uterus (6), and rectum (7).

**Figure 6 medicina-59-00072-f006:**
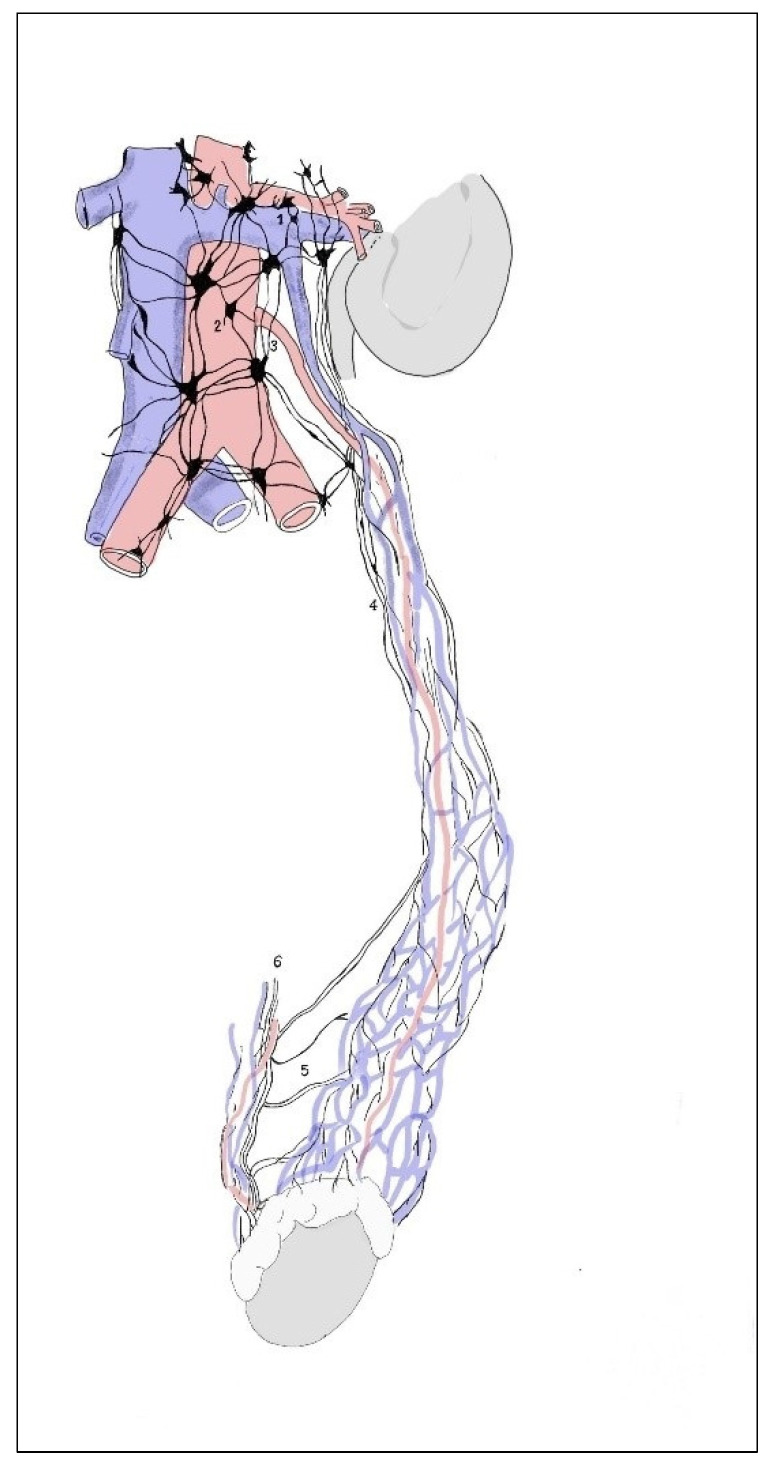
Differences in the structure of the spermatic plexus. In the segmental type, they consist of numerous small ganglia and nerves. They encircle the spermatic vasa in the form of networks, and the renal plexuses (1), aortic plexuses (2), mesenteric plexus (3), and other plexuses consist of many small nodules and nerves connected to each other; they surround the vasa spermatica and arise from the above-mentioned plexuses. In the course of the nerves (in their proximal part), it is possible to observe a large number of small ganglia (4). In the periphery of such a structure, one can observe many connections (5) with the deferential plexus (6).

**Figure 7 medicina-59-00072-f007:**
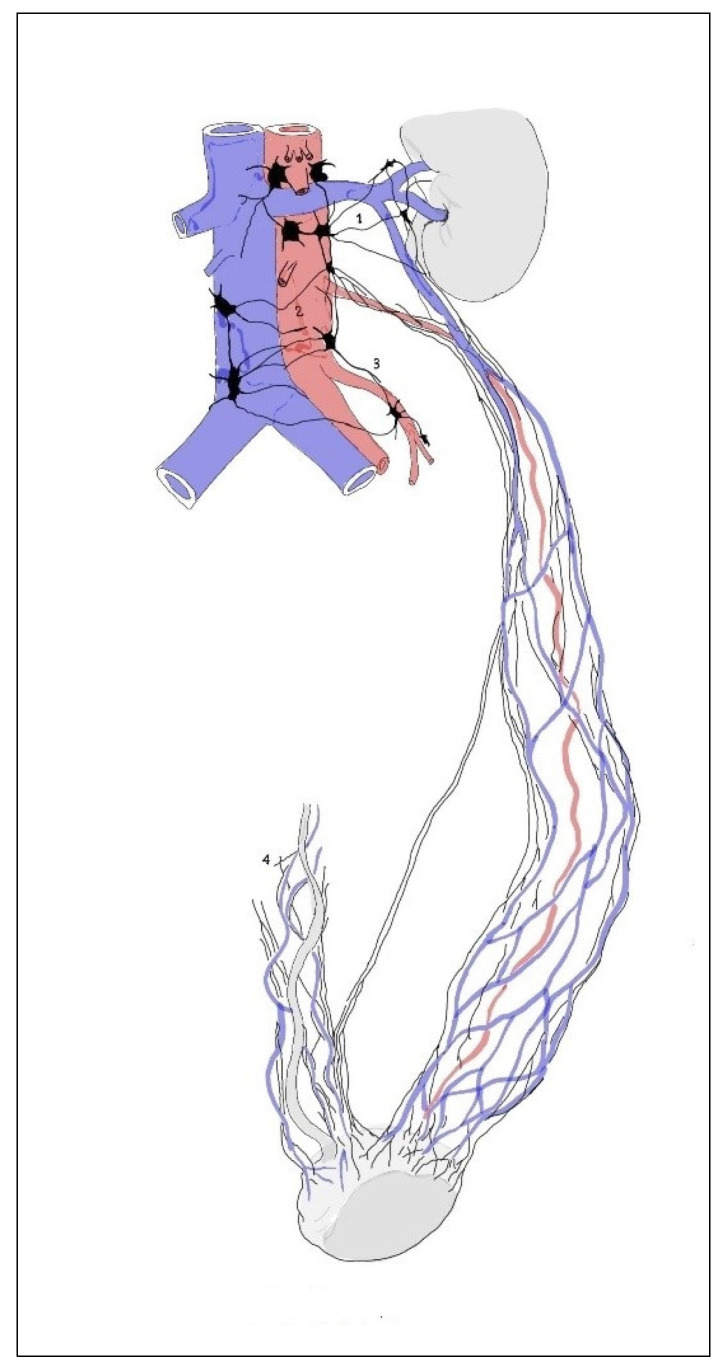
In the concentric type, the ganglionic masses are concentrated in a smaller number of large nodes. Connections between the plexus nerves on the periphery are less common. When the ganglionic masses are concentrated in a smaller number of large nodes, forming the renal plexus (1), aortic plexus (2), and mesenteric plexus (3), the spermatic plexus consists of a smaller number of thin nerves arising mainly from the renal and aortic plexuses. Between the nerves at the periphery of the plexus, the connections are less common. In similar cases, the connections of the spermatic plexus with the deferential plexus (4) are also rare.

**Figure 8 medicina-59-00072-f008:**
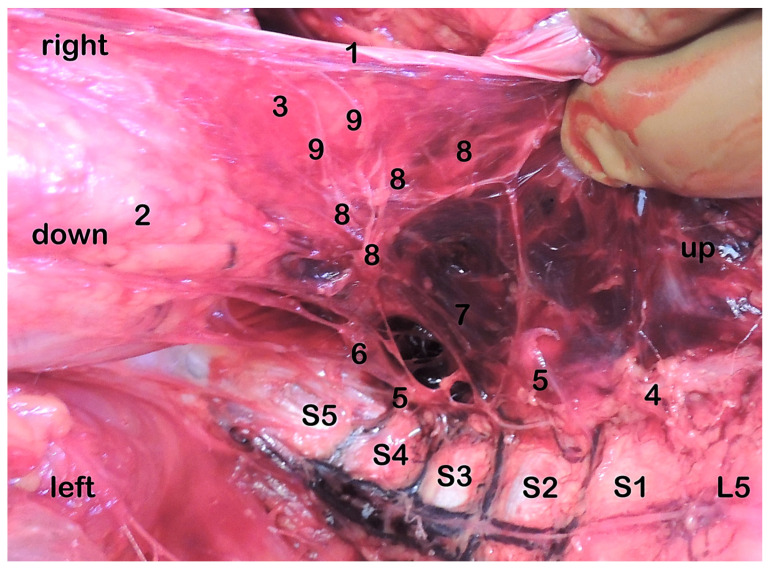
Pelvic plexus, front view, craniocaudal direction. Significant asymmetry of the pelvic plexus is observed. Peritoneum (1), rectum (2), bladder (3), truncus sympathicus (4), ganglia trunci sympathici (5), postganglionic sympathetic branches (6), pelvic splanchnic nerves (7), ganglia plexus pelvicus (8). Subperitoneal penetration of the branches of the plexus to the organs of the lesser pelvis (9).

**Figure 9 medicina-59-00072-f009:**
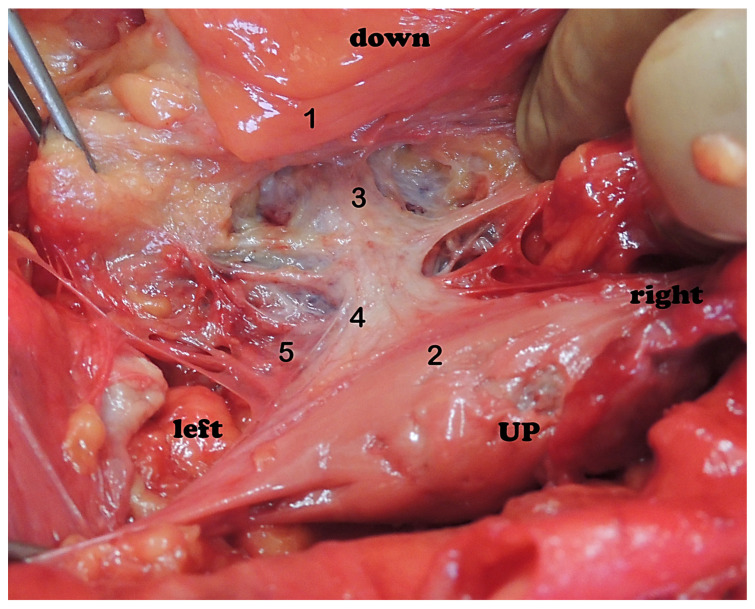
Uterovaginal plexus, front view, craniocaudal direction. Asymmetric formation of the uterovaginal plexus (5), uterus (2), vagina (3), small ganglia on the anterior and lateral sides of the vagina (4), bladder (1).

**Table 1 medicina-59-00072-t001:** Division of cases according to lateral asymmetry of the pelvic plexus.

Men (17)		Women (3)	
Right	Left	Right	Left
5	12	1	2

**Table 2 medicina-59-00072-t002:** Division according to the type of formation of peripheral ANS.

Men	Women	Segmental Type	Concentric Type	Symptoms of Diffuse Form
17	3	14	6	20

## Data Availability

Not applicable.

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
