# Peer review of "Morphological Peculiarities of the Pelvic Autonomic Nervous System and Their Impact on Clinical Interventions in the Lesser Pelvic Region"

_medicina, 2022, doi:10.3390/medicina59010072_

Round 1

Reviewer 1 Report

I really enjoyed editing this well-written article. The edits I recommend are minor. My only apprehension with this article is the cadaver population. The author states that these bodies were victims of violent deaths. I understand the standards for donation may be different in Slovakia, but it is normal for donors to be the victims of violent deaths? This would not occur in the US, for instance, so I felt my comment is needed to be mentioned for its ethical concerns. I also understand that it is mentioned right after that the ethics committee approved this study, but I’m wondering about the standards of that committee.

Author Response

We appreciate the time and effort that you have dedicated to providing your valuable feedback on our manuscript. We have been able to incorporate changes to reflect most of the suggestions provided by the academic editor and the reviewers. We have highlighted the changes within the manuscript (green color).

  • The author states that these bodies were victims of violent deaths. I understand the standards for donation may be different in Slovakia, but it is normal for donors to be the victims of violent deaths? This would not occur in the US, for instance, so I felt my comment is needed to be mentioned for its ethical concerns. I also understand that it is mentioned right after that the ethics committee approved this study, but I’m wondering about the standards of that committee.

Response: Subjects died by violent death – from medical – forensic point of view. Subjects died most often in car accidents, or by sudden death and we added to the text also death due to asphyxia. So they were not victims of homicides. Also, it was not about donation, but about research on cadavers.

The authors also attached the statement of the Ethics committee of the Health Care Surveillance Authority with the specific number.

Reviewer 2 Report

For Author

This submitted paper was a original report titled "Morphological peculiarities of the pelvic autonomic nervous system and their impact on the clinic in the small pelvis region". The authors tried to discuss stat of various testicular cells in point of "Variation of pelvic autonomic nervous system".

This study is of great interest in pelvic surgery. However, there are quite a few problems with the content of the thesis (including how to write it). The reviewer will give some comments as described below.

L15: morphological peculiarities

1) For morphological characteristics, the author should be specific.

L18: 20 cadavers

2) The author should indicate the number of all corpses. Are the 20 cadavers all that the author dissected? ?

L43: Introduction

3) The author should give a general ANS definition.

4) The author should give a general IHP definition.

5) The author should show previous anatomical reports.

6) The author should show something that has not been understood so far.

7) The author should be specific about their purpose.

L61: Deviations in the formation of the pelvic plexus were observed in 20 cadavers.

8) The author should indicate the number of all corpses. Are the 20 cadavers all that the author dissected? ?

9) The author should indicate the fixation method.

L64: The study was approved by the ethics committee.

10) The author should give a specific number.

11) Is it the same as the Ethics Committee Health Care Surveillance Authority? ?

L74: By pressing on the lower pole of the scrotum, we pushed the testis and epididymis upwards and visualized the plexus spermaticus and plexus deferentialis.

12) The author should clearly indicate the method of dissection of the inguinal canal.

L85: segmental type

13) The author should give a concrete explanation.

14) Actual typical photographs of this nerve runs are required.

L86: When concentrated in 6 (30%) (Tab. 1).

15) The author should give a concrete explanation.

16) Actual typical photographs of this nerve runs are required.

L87: the diffuse type

17) The author should give a concrete explanation.

18) Actual typical photographs of this nerve runs are required.

L93: rr. interganglionares

19) The author should give a concrete explanation.

L94: In the sacral section, rr. communicantes and rr. interganglionares were more often observed with cellular inclusions (small ganglia) (Fig. 2) in 5 (25%) cases in the concentrated type, we observed a decrease in the number of ganglia tr. sympathicus, (Fig. 3) as well as the absence of small ganglia on the connecting branches.

20) The author should also explain the remaining 15 examples.

21) Actual typical photographs of this nerve runs are required.

L123: In the small pelvis, we observed a relationship between the dimensions and the number of ganglia in segmental and concentrated types of pelvic plexus, as well as the number of nerve branches and anastomoses. In all cases there was significant lateral asymmetry.

22) The author should indicate the value.

L131: In the concentrated type (Fig. 3) the plexuses of the (1) and bladder (2) consisted of large nerve ganglia and a junction with tr. sympathicus (3) were expressed weaker than in the previous case (Fig. 1). Nn. erigentes pelvici were rare. Plexus hypogastricus inferior was formed by a small number of branches, the connections between them were innumerable. The other pictures show the uterovaginal plexus segmental type (Fig. 4) and concentrated type (Fig. 5), plexus spermaticus segmental type (Fig. 6) and concentrated type (Fig.7).

24) Why doesn't the author give a detailed explanation? ?

L165: This study is the first complete description of morphological peculiarities with relation to the clinic of pelvic ANS in fresh cadavers.

25) Reviewers were searched on PUBMED. The keyword is "pelvic autonomic nervous system cadaver". The result was 103 hits. What is FIRST? ? The author should give a concrete explanation.

L166: ANS in fresh cadavers

26) Reviewers don't understand.

L189: Not a single case met the textbook definition of normality.

27) The author should explain the textbook nerve run.

L191: Variations of ANS have only been rarely described in the past. We did not come across works dealing with pelvic ANS variations bilaterally, only with isolated papers dealing with the variations of individual ANS nerves [11, 12, 13, 17, 18, and 20]. In our group, we observed them in all cases of the lumbosacral region.

28) Reviewers were searched on PUBMED. The keyword is "pelvic autonomic nervous system cadaver variation". The result was 9 hits. Aren't these papers discussing "variation"? ?

29) “There is no difference between this study and these publications,” the author noted. What is "NEW" in this research? ?

30) Female ANS variations have already been reported. Is there any difference?

Aurore V, Röthlisberger R, Boemke N, Hlushchuk R, Bangerter H, Bergmann M, Imboden S, Mueller MD, Eppler E, Djonov V. Anatomy of the female pelvic nerves: a macroscopic study of the hypogastric plexus and their relations and variations. J Anat. 2020 Sep;237(3):487-494. doi: 10.1111/joa.13206.

Author Response

We appreciate the time and effort that you have dedicated to providing your valuable feedback on our manuscript. We have been able to incorporate changes to reflect most of the suggestions provided by the academic editor and the reviewers. We have highlighted the changes within the manuscript (green color).

Here is a point-by-point response to the comments and concerns.

L15: morphological peculiarities

1) For morphological characteristics, the author should be specific.

 Response: The authors describe morphological peculiarities in terms of anatomical variations of the pelvic ANS.

L18: 20 cadavers

2) The author should indicate the number of all corpses. Are the 20 cadavers all that the author dissected?

Response: Yes, the total number of corpses in this study was 20.

L43: Introduction

3) The author should give a general ANS definition.

Response: The authors have added the definition of ANS to the text.

The autonomic nervous system (ANS) is a division of the peripheral nervous system, the system of nerves and ganglia that innervates the blood vessels, heart, smooth muscles, viscera, and glands and controls their involuntary functions, consisting of sympathetic and parasympathetic portions.

4) The author should give a general IHP definition.

Response: The authors have added the definition of IHP to the text.

The inferior hypogastric plexus is a paired plexus of autonomic nerves ad ganglia that supplies the viscera of the pelvic cavity.

5) The author should show previous anatomical reports.

Response: The authors have added the references.

6) The author should show something that has not been understood so far.

Response: The authors have supplemented the text in the Introduction section.

7) The author should be specific about their purpose.

Response: The authors defined the aim of the work in the Abstract (first sentence) and at the end of the Introduction.

L61: Deviations in the formation of the pelvic plexus were observed in 20 cadavers.

8) The author should indicate the number of all corpses. Are the 20 cadavers all that the author dissected?

Response: Yes, the total number of corpses in this study was 20.

9) The author should indicate the fixation method.

Response: These were „fresh“, unfixed bodies (cadavers). Autopsies of these bodies were performed within 24 hours of death.

L64: The study was approved by the ethics committee.

10) The author should give a specific number.

Response: We have noticed the missing approval code in the document and included it.

The authors also attached the statement of the Ethics committee of the Health Care Surveillance Authority with the specific number.

11) Is it the same as the Ethics Committee Health Care Surveillance Authority?

Response: Yes, it is.

L74: By pressing on the lower pole of the scrotum, we pushed the testis and epididymis upwards and visualized the plexus spermaticus and plexus deferentialis.

12) The author should clearly indicate the method of dissection of the inguinal canal.

Response: The inguinal canal was cut with a short incision. This was followed by the procedure described by the authors in the text.

L85: segmental type

13) The author should give a concrete explanation.

Response: According to the number of ganglia, the variations of pelvic ANS were classified as segmental type (3 or more) and concentrated type (2 or less). We added it to the text.

14) Actual typical photographs of this nerve runs are required.

Response: Due to the fact that the research involved fresh cadavers, the macroscopic photos are not clear enough (unequivocally) – that´s why drawings were used.

The authors included 2 photos in the article (Fig. 8 and 9) showing the asymmetry of the pelvic plexus ANS.

L86: When concentrated in 6 (30%) (Tab. 1).

15) The author should give a concrete explanation.

Response: According to the number of ganglia, the variations of pelvic ANS were classified as segmental type (3 or more) and concentrated type (2 or less). We added it to the text.

16) Actual typical photographs of this nerve runs are required.

Response: Due to the fact that the research involved fresh cadavers, the macroscopic photos are not clear enough (unequivocally) – that´s why drawings were used.

L87: the diffuse type

17) The author should give a concrete explanation.

Response: The diffuse type is a mixed type, as the authors stated in the text - With the diffuse type, we observe transitions between both types of ANS formation.

18) Actual typical photographs of this nerve runs are required.

Response: Due to the fact that the research involved fresh cadavers, the macroscopic photos are not clear enough (unequivocally) – that´s why drawings were used.

L93: rr. interganglionares

19) The author should give a concrete explanation.

Response: The authors added an explanation to the text.

L94: In the sacral section, rr. communicantes and rr. interganglionares were more often observed with cellular inclusions (small ganglia) (Fig. 2) in 5 (25%) cases in the concentrated type, we observed a decrease in the number of ganglia tr. sympathicus, (Fig. 3) as well as the absence of small ganglia on the connecting branches.

20) The author should also explain the remaining 15 examples.

Response: The authors have added it to the text. At the same time, they corrected a numerical mistake in the text – instead of 5 by 6 concentrated type; 14 segmental type.

21) Actual typical photographs of this nerve runs are required.

Response: Due to the fact that the research involved fresh cadavers, the macroscopic photos are not clear enough (unequivocally) – that´s why drawings were used.

L123: In the small pelvis, we observed a relationship between the dimensions and the number of ganglia in segmental and concentrated types of pelvic plexus, as well as the number of nerve branches and anastomoses. In all cases there was significant lateral asymmetry.

22) The author should indicate the value.

Response: Asymmetry was observed in all cases. The authors added to the text the distribution of variant types according to gender. They also added the distribution according to asymmetry – Table 2.

L131: In the concentrated type (Fig. 3) the plexuses of the (1) and bladder (2) consisted of large nerve ganglia and a junction with tr. sympathicus (3) were expressed weaker than in the previous case (Fig. 1). Nn. erigentes pelvici were rare. Plexus hypogastricus inferior was formed by a small number of branches, the connections between them were innumerable. The other pictures show the uterovaginal plexus segmental type (Fig. 4) and concentrated type (Fig. 5), plexus spermaticus segmental type (Fig. 6) and concentrated type (Fig.7).

24) Why doesn't the author give a detailed explanation?

Response: These were anatomical variations that the authors observed in individual cases.

L165: This study is the first complete description of morphological peculiarities with relation to the clinic of pelvic ANS in fresh cadavers.

25) Reviewers were searched on PUBMED. The keyword is "pelvic autonomic nervous system cadaver". The result was 103 hits. What is FIRST? The author should give a concrete explanation.

Response: The „FIRST“ is that the subjects were fresh cadavers. Previous studies have been conducted on preserved cadavers. At the same time, the study deals with the ANS bilaterally and divides the anatomical variations of the pelvic ANS into segmental and concentrated types.

L166: ANS in fresh cadavers

26) Reviewers don't understand.

Response: These were fresh cadavers (bodies), in the meaning that they were not preserved, embalmed or fixed in any other way. Autopsies and studies were performed on the bodies within 24 hours of death.

L189: Not a single case met the textbook definition of normality.

27) The author should explain the textbook nerve run.

Response: The anatomical definition of „normality“ is given in several anatomy textbooks. In this article, the authors used the definitions from the supplemented reference no. 1.

  • Kachlík D, Varga I, Báča V, Musil V. Variant Anatomy and Its Terminology. Medicina (Kaunas). 2020; 56 (12): 713. Published 2020 Dec 18. doi:10.3390/medicina56120713.)

L191: Variations of ANS have only been rarely described in the past. We did not come across works dealing with pelvic ANS variations bilaterally, only with isolated papers dealing with the variations of individual ANS nerves [11, 12, 13, 17, 18, and 20]. In our group, we observed them in all cases of the lumbosacral region.

28) Reviewers were searched on PUBMED. The keyword is "pelvic autonomic nervous system cadaver variation". The result was 9 hits. Aren't these papers discussing "variation"?

Response: The authors modified the text. At the same time, they added references.

29) “There is no difference between this study and these publications,” the author noted. What is "NEW" in this research?

Response: In this research, it is new that fresh unfixed bodies (cadavers) were used in which ANS dissection is difficult. Furthermore, it was a study dealing with the ANS bilaterally and the division of the detected anatomical variations of the ANS into concentrated and segmental type.

30) Female ANS variations have already been reported. Is there any difference?

Aurore V, Röthlisberger R, Boemke N, Hlushchuk R, Bangerter H, Bergmann M, Imboden S, Mueller MD, Eppler E, Djonov V. Anatomy of the female pelvic nerves: a macroscopic study of the hypogastric plexus and their relations and variations. J Anat. 2020 Sep;237(3):487-494. doi: 10.1111/joa.13206.

Response: This study deals with the ANS bilaterally and divides the identified anatomical variations of the ANS int concentrated and segmental type. The authors conclude that their study is affected by the small number of observations, especially regarding women.
